# Factors influencing the E-learning system usage during the COVID-19 pandemic in Vietnam

**Thang Xuân Do**\*, **Huong Thi Lan Tran, Thuy Thu Le**

Department of Pharmacoeconomic and Administration, Hanoi University of Pharmacy, Hanoi, Vietnam

\* thangdx@hup.edu.vn

**Data Availability Statement:** Data are available from the following Figshare repository URL: https://figshare.com/articles/dataset/PONE-22-15672R2_xlsx/21564411 (DOI: 10.6084/m9.figshare.21564411).

## Abstract

The outbreak of the COVID-19 pandemic has brought forward an unprecedented situation, which has forced the educational institutes worldwide to use a propriety "*online only*" model for teaching and learning. Teachers have been compelled to deliver lectures online using some form of an online delivery platform. In this dilemma situation with the closure of the educational institutes, one of the very basic necessities is to ensure quality features of e-learning systems that are being used for the purpose of education delivery, particularly from the students' perspective. The objective of this study was to identify factors affecting pharmacy students' satisfaction in Vietnam during the COVID 19 pandemic. A cross-sectional survey of a consecutive sample of 1612 valid responses from students who have been in distance learning at Hanoi University of Pharmacy has been conducted. An integrated model with three main dimensions had been developed: learner's characteristics; instructor's characteristics; system, and technology. Data were collected in the field on both online and offline platforms using the questionnaire of 39 items to investigate the critical factors affecting learners' satisfaction in e-learning. The reliability of the questionnaire was measured using Cronbach's alpha on this data. Descriptive statistics, factor analysis, and multiple regression analysis were employed for data analysis. Out of 2491 questionnaires distributed, 1612 questionnaires were completed (respond rate 64.7%). The results revealed that perceived usefulness, perceived ease of use, system and technical dimension and instructors' characteristics are the critical factors affecting learners' perceived satisfaction. The perceived usefulness of the students was the most important factor affecting overall satisfaction (beta coefficient = 0.610). Multiple regression analysis yielded the four main factors explaining 59.9% of total satisfaction. The findings revealed how to improve learners' satisfaction and further strengthen their e-learning implementation. The interventional solutions on students' characteristics, instructors' characteristics and system & technical dimension should be considered and implemented to improve the quality of e-learning and students' satisfaction at Hanoi University of Pharmacy.

**Funding:** The author(s) received no specific funding for this work.

**Competing interests:** The authors have declared that no competing interests exist.

## Introduction

The outbreak of the COVID-19 epidemic has led to enormous impacts on the daily lives, behavior, and awareness of Vietnamese people. The COVID-19 pandemic caused by a new strain of coronavirus (SARS-CoV-2) has affected many countries and territories worldwide and is officially recognized by the World Health Organization (WHO) declared a global pandemic [1]. The COVID-19 pandemic has disrupted teaching in many institutions, especially in pharmacy or medical schools. In many countries, including Vietnam, concentrated learning activities have had a period of pause to ensure the safety of students and lecturers [2].

To minimize disruption in the learning process, schools had to find a different approach in training students. In that context, online learning has become the core method of teaching during the COVID-19 pandemic. On March 23, the Ministry of Education and Training of Vietnam issued document No. 988/BGDĐT-GDĐH to guide higher education institutions to implement distance learning activities during the COVID-19 pandemic in order to ensure educational effectiveness and quality. Currently, most schools have successfully implemented online teaching activities through online learning channels, integrated software such as Google Meet (Google, Mountain View, CA, USA), Microsoft Team (Microsoft, Redmond, DC, USA), Zoom (Zoom Video Communications, San Jose, CA, USA)... [3]. Although online learning is the first choice in the context of lengthy and complicated translation, in fact, this process has arisen many invades and causes many difficulties for both teachers and learners due to limitations in technology infrastructure, learning conditions, or the ability to understand and follow the lesson.

Online learning, like any teaching method, has advantages and disadvantages for both students and teachers. Besides the epidemiological benefits of online learning during the COVID-19 pandemic, other benefits mentioned include increased convenience, accessibility to resources regardless of location and time [4–6]. Otherwise, e-learning initiatives also require considerable investments in technology such as hardware costs, software licenses, learning material development, equipment maintenance, and training also has many limitations, including problems of poor internet connection quality and learners' computer skills [7, 8]. Not being very familiar with such teaching methodologies, instructors and students struggle to consolidate these with their plan of continuing medical education while maintaining the quality of it. Lack of expertise in operating the electronic resources and the concomitant problem of limited access to the internet, computers, and other facilities due to social and economic setbacks is an obstacle for online learning and teaching [9]. During the Covid 19 pandemic, many studies have been conducted to evaluate the education of health students and medical specialties, however, there has not been to research evaluating the education activities of Vietnamese pharmacy students [10, 11]. Hence, this research was conducted to obtain the satisfaction of pharmacy students regarding the transition to online learning.

## Materials and methods

### Participants

Pharmacy students from the 2nd year to 5th year who joined online training at HUP during COVID 19 were recruited to participate in the study. Our research war carried out when students studied the theory and practiced directly at HUP after the online learning period due to the Covid-19 epidemic. The survey was conducted on both online and offline platforms at Hanoi University of Pharmacy in Vietnam, using a convenience sample of students. The questionnaires were administered by two different data collection methods: online questionnaire and offline questionnaire. For each class, we only did offline or online form. The offline survey

was applied for students from the 2ⁿᵈ year to 4ᵗʰ year who participated in the practical session or the theoretical session. Reseacher collected the questionnaire form in the field, so we could control their response. Student presenting at the practical session or the theoretical session were recruited to the offline. The online form was only applied to final year students who finished the theory or practice at school. When participating in the online survey, the students accessed the questionaire through the group on the online platform of their class. Students accessing and participating in answering the online questionaires would be aggregated to analysis. The personal information in the online questionaire including name, year of birth, class, phone number and email was used to check for duplicates. Data were gathered from 1277 offline answer sheets (response rate 72.1%) and 335 online answer sheets (response rate 44.6%). Whilst the final dataset is based on a convenience sample, the demographic statistics are broadly consistent with those of the population as described earlier in the section.

## Procedure and materials

Two types of questionnaires were designed for purpose of collecting suitable data from the two platforms (online survey and offline survey). The designed questionnaires were formed by the literature review which identified potential critical success factors (CSF's) and their categorization. The questionnaire included 43 items, of which 4 items were about the characteristics of the survey participants, 39 left items, which were designed on a 5-point Likert scale (1: totally disagree; 2: somewhat disagree; 3: Neither agree or disagree; 4: somewhat agree; 5: totally agree), focused on three main factors: system & technical dimension, students' characteristics, and instructors' characteristics. The questionnaire was initially developed in the English language, then a translation of the questionnaire to Vietnamese was completed. The translation was an iterative process with principles of translation/back translation. Any discrepancies were discussed and resolved. To ensure that the questionnaires were fit for the purpose, the pilot study was conducted. The pilot study used a Vietnamese language version of the questionnaire to collect data from a sample of twenty students who are currently studying at Hanoi University of Pharmacy, following which further fine adjustments were made to produce the final version.

The survey online form was designed by the Google Forms application, then the form was distributed to the students' email address which were provided by the university. All items in the survey were required to be completed. However, it was optional for respondents to provide personal information. So it allowed respondents to answer the questionnaire comfortably and to minimise bias. All the respondents' responses stored in Google Forms were exported as Master Excels (csv file) to import to SPSS 22. Before performing the analysis methods, the variables were renamed and the data were checked for the validity of each case, no cases were excluded by the respondents provided constant value for all items. The final dataset had 335 valid cases.

The survey offline was administered to 1722 students at lecture halls and laboratories based on their class schedules and practice schedules. The answer sheets were collected immediately by the research team after learners completed. After that, the researcher checked for the validity of the questionnaire (do not fill in the wrong ideas, do not choose 2 or more answers for a question), if not, it will be returned to the student to complete. The final dataset had 1277 valid cases.

## Data analysis

The information including year of birth, gender, and school year was collected to analysis and cannot track participants. The personal information including name, class, phone number and email which can track participants would be coded into ID and only one person could access the original data and this person would not participate in data analysis. The cleaned and coded

data was imported to IBM SPSS for the purpose of analysis. The data is cleaned for missing values and the questionnaires with incomplete responses were removed from the analysis. Descriptive statistics were calculated for the demographic factors to understand the basic nature of the sample considered for the study. The percentage frequencies, mean scores, and factor loadings on the questionnaire items were performed. Internal consistency reliability was assessed using Cronbach's alpha to measure the internal consistency of the scale, with preferred values higher than 0.7.

Once descriptive statistics had been generated, suitability of the dataset for Exploratory Factor Analysis (EFA) was established using the Kaiser-Meyer-Olkin (KMO) measure of sampling adequacy and Bartlett's Test of Sphericity. The selection criteria are KMO index above the acceptable minimum of 0.50, and Bartlett's test had a level of significance above the required level of 0.05%.

Factor analysis was employed to define factors and related items. A Principal Component Analysis was undertaken to find out the factors which explain the most variance in the data used. Only those factors which are having Eigenvalue above 1 are extracted. Next, orthogonal varimax rotation was used to generate a component matrix, which shows the loading of items onto the identified factors for both data sets. Items with factor loadings greater than 0.50 were considered "significant".

### Ethical issues

The study protocol was approved by the ethical committee of Hanoi University of Pharmacy, Hanoi, Vietnam (referene number 20-08/PCT-HĐĐĐ). Informed consents were obtained from all participants verbally with the witness of the class monitor. The information about participants' consent and confidentiallity were incorporated in the online questionnaire in the introduction to the purpose of the study. If consent, the participants could access to the link and answer the questionnaire.

## Results

### Demographics of the participants

The valid questionnaires were 1612 and comprised 335 online platforms and 1277 offline platforms with an overall response rate of 64.7%. As can be seen, the offline response rate was much higher than the online response rate (72% vs 46.6%). Table 1 exhibits the demographic data of the participants in terms of gender, Internet-connected devices and forms of internet connection. It shows that more than half of the participants are females (72.4%). The majority of students use phones and laptops to access (69.5% and 81.2%, respectively) with the main form of connection being wi-fi (91.8%).

**Cronbach alpha and descriptive statistics.** The Alpha Cronbach Coefficient, the reliability test, was applied to the data in order to assess the validity and the reliability of the instruments employed. Table 2 shows Cronbach's alpha for each dimension. Cronbach's alpha of three factors were higher than 0.8 (ranging from 0.890 to 0.952).

The Kaiser-Meyer-Olkin (KMO) measure of sampling adequacy for the factor analysis was 0.974. The value of Bartlett's test of Sphericity was less than 0.01 so that factor analysis may be useful for the data.

### Rotated factor loadings for current services items

Factor analysis extracted four factors with the cumulative explained variance of 59.90%. Factor loadings after a varimax rotation are shown in Table 3. In cases where the factor loading was

**Table 1. The demographic characteristics of the participants.**

|  | Frequency | Percentage |
|---|---|---|
| **Gender** |  |  |
| Male | 409 | 27.6 |
| Female | 1072 | 72.4 |
| **Internet-connected devices** |  |  |
| Mobile phone | 1029 | 69.5 |
| Laptop | 1203 | 81.2 |
| Desktop | 145 | 9.8 |
| Tablet | 66 | 4.5 |
| **Forms of internet connection** |  |  |
| Wi-Fi | 1360 | 91.8 |
| 3G/4G | 505 | 34.1 |
| **Places of connection** |  |  |
| Home | 1455 | 98.2 |
| Internet shop | 54 | 3.6 |
| **Academic Year** |  |  |
| 2 | 534 | 33.1 |
| 3 | 335 | 20.8 |
| 4 | 454 | 28.2 |
| 5 | 289 | 17.9 |
| E-learning was an appropriate solution in Covid 19 pandemic, mean (SD) | 4.39 | 0.674 |

above 0.5 for two factors and the difference between two-factor loading is less than 0.3, that observation variable would be removed. Accordingly, the three items were excluded and the 36 items on current services were regrouped into four new factors: Perceived ease of use, Perceived usefulness, Instructor's characteristics, System & technical dimension.

## Regression results

Multiple regression was employed to test the models. The findings showed in Table 4 discloses that R square = 0.599, and the p-value for the F test statistic is less than 0.0001, providing strong evidence against the null hypothesis.

Four analyzed factors exhibited significant relationships with perceived e-learner satisfaction. As listed in Table 4, among the factors influencing student satisfaction, perceived usefulness ($\beta$ = 0.610, Sig <0.005) and perceived ease of use ($\beta$ = 0.350, Sig <0.005) showed the greatest effects, following by system & technical dimension ($\beta$ = 0.241, Sig <0.005). Instructor's characteristic is evaluated at the lowest position ($\beta$ = 0.221, Sig <0.005). Through the value of adjusted R square, the explanatory level of the model was 59.9%. It showed that 59.9% of learner satisfaction can be explained by the four main factors. Based on the results, an equation is established as follow: $Y = 3.935 + 0.221^*X_1 + 0.610^*X_2 + 0.350^*X_3 + 0.241^*X_4$.

**Table 2. The Alpha Cronbach Coefficient values for each factor.**

| No. | Factor | No. of Statements | Alpha Cronbach Coefficient |
|---|---|---|---|
| 1 | System & technical dimension | 10 | 0.890 |
| 2 | Students' characteristics | 13 | 0.948 |
| 3 | Instructor's characteristics | 16 | 0.952 |

**Table 3. Final factor loading.**

| Factors | Item | Component |
|---|---|---|
| **Instructor characteristics** | Instructor's style of teaching using e-learning technologies | 0.761 |
| | Instructor's enthusiasm while teaching using e-learning tools | 0.733 |
| | Students were invited to ask questions/receive answers | 0.721 |
| | The instructor's style of presentation holds my interest | 0.720 |
| | The instructor is friendly towards individual students | 0.715 |
| | The instructor is active in teaching me the course subjects via e-learning | 0.714 |
| | The instructor explains how to use the e-learning components | 0.714 |
| | The instructors impart easy-to-understand knowledge suitable for e-learning | 0.704 |
| | Students felt welcome in seeking advice/help | 0.704 |
| | The instructors have a method of assessing students' participation and knowledge acquisition | 0.701 |
| | The instructors answer questions from students in a timely manner when teaching online | 0.671 |
| | Instructor's ability to use the e-learning system effectively | 0.652 |
| **Perceived Usefulness** | My attention to the class tasks during e-learning session was greater in comparison to the traditional face-to-face class meetings. | 0.848 |
| | The activities during the e-learning sessions motivated me to learn the class content more than the ones in the traditional face-to-face class meetings. | 0.831 |
| | The use of e-learning improved my learning in the class. | 0.791 |
| | Using the e-learning system will improve my learning performance | 0.729 |
| | The use of e-learning motivated me to seek help from tutors, classmates, and the instructor. | 0.721 |
| | Using the e-learning system will make it easier to learn course content | 0.692 |
| | I am more comfortable responding to questions by email than orally | 0.685 |
| | I find the e-learning system useful in my learning | 0.626 |
| | My technical skills (email/internet apps) have increased since attending online classes | 0.610 |
| | I prefer my online courses as they are very structured with set due dates similar to face-to-face courses | 0.580 |
| **Perceived Ease of Use** | I could complete my learning activities using the web-based system even if there is no one around to show me how to do it | 0.781 |
| | I find the e-learning system easy to use | 0.747 |
| | I could complete my learning activities using the web-based system even if I had never used a system like it before | 0.732 |
| | I could complete my learning activities using the web-based system If I had only the system manuals for reference | 0.724 |
| | It is easy for me to become skillful at using the e-learning | 0.723 |
| | I have no difficulty when operating on e-learning devices | 0.649 |
| **System & technical dimension** | Learning materials are easy to download (Download) | 0.681 |
| | The e-learning system offers multimedia (audio, video, and text) types of course content | 0.646 |
| | E-learning software is compatible with electronic devices (phones, computers, etc.) | 0.644 |
| | The e-learning system enables interactive communication between instructor and students | 0.609 |
| | E-learning system that allows students to record lectures | 0.599 |
| | The course materials were placed online in a timely manner | 0.587 |
| | Teaching content is useful to students | 0.579 |
| | Overall, the e-learning system was easy to use | 0.553 |

## Discussion

To our knowledge, the current study is among the first to address pharmacy student distance online learning and associated factors in Vietnam during Covid 19 pandemic. Some previous studies indicated the distance education of pharmacy student during the COVID 19 pandemic [12, 13]. The Covid 19 pandemic affected the pharmacy education so that adjustments are needed to accommodate the education in the new situation. Online training is considered a suitale form for low and middle-income countries like Vietnam [14]. The form of e-learning training has shown its effectiveness in pharmacy training [15].

**Table 4. Coefficients.**

| Factor | Unstandardized Coefficients | Standardized Coefficients | Sig. | VIF |
|---|---|---|---|---|
| Constant | 3.935 | - | <0.0001 | 1.000 |
| Instructor's characteristics (X1) | 0.195 | 0.221 | <0.000 | 1.000 |
| Perceived usefulness (X2) | 0.537 | 0.610 | <0.0001 | 1.000 |
| Perceived ease of use (X3) | 0.308 | 0.350 | <0.0001 | 1.000 |
| System & technical dimension (X4) | 0.212 | 0.241 | <0.0001 | 1.000 |
| Sig.F | <0.0001 | | | |
| Adjusted R square | 0.599 | | | |

The results relate to other precedent CSF studies. Abdulla (2018) identified the following four categories: students' characteristics, teachers' characteristics, technology, and design and content [16]. Selim (2007) identified seven factors, with three focusing on student characteristics, and the other four being instructors' characteristics, technology, support, and e-learning system [17]. There are several potential reasons for this, including differences in respondent's background, curricula, culture, facilities, or the items used in the instrument. In addition, where there are some similarities in categories, there remain differences in the relative ranking of CSFs.

## Perceived usefulness

The research findings indicated perceived usefulness as one of the key factors that must be considered and planned for before the implementation of e-learning. Participants rates their motivation and concentration with low scores (3.43 and 3.55, respectively). This is because online learning requires learners with high self-discipline. The learning environment outside the lecture hall is easy for learners to be distracted. Besides this, studying for a long time continuously makes students face many health problems such as headaches, tinnitus. These are also the obstacles learners face when switching from traditional to online learning.

The positive association between perceived usefulness and user satisfaction is consistent with previous studies. Perceived usefulness is also found to be a significant factor affect e-learning usage intention which confirms the studies carried out by Motaghian (2013) [18], Chen and Tseng (2012) [19], Chow (2012) [20], Li (2012) [21] and Sumak (2011) [22]. It can be seen that the perceived usefulness of an e-learning system is an important external driver for user satisfaction with the system. Hence, the higher the perceived usefulness of an e-learning system, the more satisfaction learners had.

## Perceived ease of use

It was found that the second factor that affects e-learners' satisfaction is perceived ease of use. Students indicated that the systems are easy to use and manipulate. Perceived ease of use helps to increase student motivation and attitude towards online learning. These foster student-faculty interaction, encourage learners to develop study skills and broaden the scope of the learning experience. The ease of use of the E-Learning system helps individuals to achieve better learning efficiency, thus, the level of learning satisfaction will be higher.

Chen (2012) [19] and Ong (2004) et al [23] concluded that e-learning acceptance is influence directly by perceived usefulness and indirectly by perceived ease of use. User-friendly design is an important factor affecting the success of e-learning implementation. Ease of use contributes to increased student motivation and attitude towards online learning, which encourages learners to develop study skills and promotes student-faculty interaction.

## System & technical dimension

In the literature, Selim et al [17] identified system & technical dimension as a significant factor influencing the implementation of e-learning. This research finding shows that system & technical dimension represents an important factor during the implementation of e-Learning.

These variables *"Learning materials are easy to download"*, *"Microsoft-Team online learning software compatible with electronic devices"*, *"Teaching content useful to students"* were rated at the same level with an average score around 4.1. The result reveals the importance of learning material and content. Consistent with other studies, the topic of the online activity was the important factors in deciding upon participation [11]. Therefore, the design of learning materials and content is an important factor when implementing online training.

System functionality and system interactivity directly influence e-learner satisfaction. However, factors related to the system's functionality including lecture recording, materials posted online promptly were rated at low score. The online learning process is easily interrupted due to the quality of the transmission or problems on the device. Therefore, it is necessary to allow recording of lectures during the learning process to help learners review more conveniently.

The system function is also one of the key factors affecting student satisfaction when using the system. Learners find that the system is equipped with the right functionality that increases perceived ease of use and perceived effectiveness, and thereby increases their satisfaction. Besides, the system functional specification also suggests specific characteristics that are the target for e-learning system developers should aim for. For example, in this study, learners found that the system needed to add recording functionality and allow better remote access to course content. In Selim's research, learners have shown that a system function that helps learners and teachers interact effectively will increase learner satisfaction [17]. Therefore, besides building the right system interface, the system's feature characteristics including enabling efficient interactions and providing access to course content and recording also play an important role in influencing student satisfaction.

## Instructor's dimensions

The impacts of teachers' characteristics on e-learning implementation were largely influenced by the teachers' attitudes, control of technology in the learning environment, and teaching style. These variables "*The instructors answer questions from students in a timely manner when teaching online*" and "*Instructor's enthusiasm while teaching using e-learning tools* "are rating at the highest score nearly 4.2. The majority of the participants agreed that the teachers' attitude factor is one of the most significant factors during e-learning implementation. One possible explanation of this finding is as follows: teachers play a vital role in the learning process in general and in e-learning in particular. The more enthusiastic teachers are about e-learning, the more they will motivate students in all their educational practices. While rapid transition to distance online elearning was a mandatory and necessary action to enssure the learning continuity during the COVID 19 pandemic, that posed many chanllenges for instructors. However, educational institutions can help their teachers adapt to new forms of education [24].

Ali et al (2011) [25] indicated that the teacher's behavior affects the interaction between students and lecturers as well as the acceptance and satisfaction of students with the learning experience. Besides, the research results also show that the willingness of lecturers to consult, provide feedback and answer questions (especially in distance education) is very important, having an impact on students learning experiences [25]. This result is also consistent with the studies of Selim (2007) [17] and McPherson and Nunest (2008) [26], indicating that the teaching style and method are applied in the teaching process, and the cooperation and interaction

in the teaching process. The teaching process also increases the "motivation and attitude towards e-learning" of students and improves student satisfaction.

Several limitation should also be considered. First, the collection data was conducted when students of HUP learnt directly at school and the online learning stopped. We only conducted online surveys through submitting a google form on the general class groups and could not attached on the e-learning platform. Therefore, not every students in the class had access to our online questionnaires. This may lead to the low online response rate. The direct survey form was preferred to increase the response rate. Second, due to the nature of a cross-sectional design, we cannot establish causal relationships between independent variables and the satisfaction. Third, the generalization of our study is limited by the convenience sampling strategy.

## Conclusion

This study revealed critical success factors influencing e-learners' satisfaction. It can be concluded that instructor dimension, perceived ease of use, perceived usefulness, and system and technological dimensions, are the most important success factor dimensions to influences the satisfaction of e-learner. This study provides insights for institutions to strengthen their e-learning implementations and further improve learner satisfaction. Thus, institutions may be recommended that they should pay more attention to the aforementioned factors to ensure the successful implementation of an e-learning system. Valuable lesson learned from E-learning period (Covid-19) will help HUP develop their distance learning programs to meet the learners' demand in some diversity situation.

## Acknowledgments

The authors would like to acknowledge the contribution of the following individuals for their assistance in developing the satisfaction questionnaire and participants in this study. We also thank to Hanoi University of Pharmacy for allowing us to conduct this research.

## Author Contributions

**Conceptualization:** Thang Xuân Do, Thuy Thu Le.

**Data curation:** Huong Thi Lan Tran.

**Formal analysis:** Thang Xuân Do, Huong Thi Lan Tran.

**Investigation:** Thang Xuân Do, Huong Thi Lan Tran.

**Methodology:** Thang Xuân Do, Huong Thi Lan Tran, Thuy Thu Le.

**Supervision:** Thang Xuân Do.

**Writing – original draft:** Thang Xuân Do, Huong Thi Lan Tran, Thuy Thu Le.

**Writing – review & editing:** Thuy Thu Le.

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
