## [Decision Letter · Decision Letter 0]

18 Jul 2022

PONE-D-22-15672Factors influencing the E-learning system usage during the COVID-19 pandemic in VietnamPLOS ONE

Dear Dr. Thang,

Thank you for submitting your manuscript to PLOS ONE. After careful consideration, we feel that it has merit but does not fully meet PLOS ONE’s publication criteria as it currently stands. Therefore, we invite you to submit a revised version of the manuscript that addresses the points raised during the review process. Kindly respond as much as possible to all the issues raised by both reviewers to improve the quality of your paper.  Highlight all significant limitations in the appropriate sections. Ensure that you review PLOS ONE author guidelines and edit your manuscript accordingly. 

We look forward to receiving your revised manuscript.

Kind regards,

Ogochukwu Chinedum Okoye

Academic Editor

PLOS ONE

Journal Requirements:

2. PLOS ONE does not copy edit accepted manuscripts (https://journals.plos.org/plosone/s/criteria-for-publication#loc-5). To that effect, please ensure that your submission is free of typos and grammatical errors. 

6. Please include a copy of Table 5 which you refer to in your text on page 8.

7. Please include your tables as part of your main manuscript and remove the individual files. Please note that supplementary tables should be uploaded as separate "supporting information" files.

Reviewers' comments:

Reviewer's Responses to Questions

**Comments to the Author**

1. Is the manuscript technically sound, and do the data support the conclusions?

Reviewer #1: Partly

Reviewer #2: Yes

2. Has the statistical analysis been performed appropriately and rigorously? 

Reviewer #1: Yes

Reviewer #2: Yes

3. Have the authors made all data underlying the findings in their manuscript fully available?

Reviewer #1: No

Reviewer #2: Yes

4. Is the manuscript presented in an intelligible fashion and written in standard English?

Reviewer #1: Yes

Reviewer #2: Yes

5. Review Comments to the Author

Reviewer #1: The research uses basic statistics.

The research focuses on one unique department of one university.

The research misses recent literature review.

The research discussion is well stablished.

The conclusions need further development.

Reviewer #2: Introduction

Line 48: Please list software properly - e.g. Zoom (Zoom Video Communications, San Jose, CA, USA)

Most points should be simply reported in Into and then discussed in the Discussion Section

Lines 63-65: “many studies have been conducted to evaluate the education of health students” would be more broadly described as “many studies have been conducted to evaluate the education of health students and medical specialties”. At this point you can cite a recent international multicenter research work, such as Vascular e-Learning During the COVID-19 Pandemic: The EL-COVID Survey, Ann Vasc Surg 2021; 77: 63–70, DOI: 10.1016/j.avsg.2021.08.001

M&M

Line 72: What does “a convenience sample” mean?

How did you make sure that participants did not submit an online and an offline survey form?

Were the questionnaires validated?

How did you validate the online submitted data? Were there any duplicates?

Line 100-104: Do you think that the filtering of the offline forms by the academic staff could result into any bias?

I was left with the impression that classic teaching was halted due to COVID. You mention that offline forms were distributed in classes and laboratories. There is a conflict in these statements. Please explain further.

Apart from email distribution of the forms, did you include an online form within the e-learning platform?

Results:

Lines 127-128: The online participation was lower than the offline participation. Wouldn’t this be strange when we are evaluating e-learning?

Discussion

Please include any segment necessary as per the above mentioned points.

Overall

Please keep a consistent reference style.

6. PLOS authors have the option to publish the peer review history of their article (what does this mean?). If published, this will include your full peer review and any attached files.

Reviewer #1: No

Reviewer #2: No

---

## [Author Response · Author response to Decision Letter 0]

20 Sep 2022

August 25th, 2022

Subject: Revision submission (PONE-D-22-15672)

Dear Prof. Ogachukwu Chinedum Okoye,

We would like to send the revision of the article entitled "Factors influencing the E-learning system usage during the COVID019 pandemic in Vietnam" for publication in Journal of Plos One. We would like to thank the reviewer(s) and editor(s) for the valuable comments and suggestion. All the issues concerned were well revised and addressed:

Reviewers' comments:

Reviewer 1:

1. The research uses basic statistics. The research focuses one unique department of one university.

Author's response: At the time of the study, my university only specialized in Pharmacy. Therefore, the study was carried out throughout the school with all pharmacy students.

2. The research misses recent literature review.

Author's response: We have updated recent studies (line 69, line 234; line 266 ); 

3. The research discussion is well stablished.

Authors' response: Thank you for your positive remarks on our manuscript. 

4. The conclusions need further development

Authors' response: We have developed more conclusions. Valuable lesson learned from E-learning period (Covid-19) will help HUP develop their distance learning programs to meet the learners’ demand in some diversity situation (line 292-294)

Reviewer 2:

5. Line 48: Please list software properly - e.g. Zoom (Zoom Video Communications, San Jone, CA, USA)

Authors' response: Thank you for your comment. We have revised the software listing: Google Meet (Google, Mountain View, CA, USA), Microsoft Team (Microsoft, Redmond, DC, USA), Zoom (Zoom Video Communications, San Jose, CA, USA) (line 48-49)

6. Most points should be simply reported in Intro and then discussed in the Discusion section

Authors' response: Thank you for your comment. We have reviewed and edited.

7. Line 63-65: “many studies have been conducted to evaluate the education of health students” would be more broadly described as “many studies have been conducted to evaluate the education of health students and medical specialties”. At this point you can cite a recent international multicenter research work, such as Vascular e-Learning During the COVID-19 Pandemic: The EL-COVID Survey, Ann Vasc Surg 2021; 77: 63–70, DOI: 10.1016/j.avsg.2021.08.001

Authors' response: Thank you for your detailed comment. We have edited and added citations (line 67 and line 69)

8. Line 72: What does “a convenience sample” mean?

Authors' response: We have added in the research method more clearly how to select the sample. Students presenting at the practical session or the theoretical session were recruited to the offline survey. The online form was only applied to final year students who finished the theory or practice at school. When participating in the online survey, the students accessed the questionaire through the group on the online platform of their class. Students accessing and participating in answering the online questionaires would be aggregated to analysis. (line 82-87). 

9. How did you make sure that participants did not submit an online and an offline survey form?

Authors' response: We have added more information in the research method. For each class, we only did offline or online form. The offline survey was applied for students from the 2nd year to 4th year who participated in the practical session or the theoretical session. Researcher collected the questionnaire form in the field, so we would control their response. (Line 79-82)

10. Were the questionnaires validated?

Authors' response: We have added more information in the research method how to validate the questionnaires. (Line 103-110)

11. How did you validate the online submitted data? Were there any duplicates?

Authors' response: We have added more information in the research method. The personal information in the online questionaire including name, year of birth, class, phone number and email was used to check for duplicates (Line 86-88). In addition, the online data was collected from only final year students, so there were not any duplicates.

12. Line 100-104: Do you think that the filtering of the offline forms by the academic staff could result into any bias?

Authors' response: The questionnaires were distributed and collected by the reseach team. Therefore, we think that no questionnaires were filtered by the academic staff. We have added this information in the research method (Line 81-82 and 119).

13. I was left with the impression that classic teaching was halted due to COVID. You mention that offline forms were distributed in classes and laboratories. The is a conflict in these statements. Please explain further

Authors' response: Due to the impact of the COVID pandemic, the online learning was applied for theoretical session at HUP in a period of time. However, when the pandemic was controlled, many Vietnamese university including HUP allowed students to return to school to learn directly. Then, the practical sessions were applied. Our research was carried out when students studied the theory and practiced directly at HUP (line 74-76)

14. Apart from email distribution of the forms, did you include an online form within the e-learning platform

Authors' response: The collection data was conducted when students of HUP learnt directly at the school (the online learning stopped). Therefore, we only conducted online surveys through submitting a google form on the general class group and could not attach on the e-learning platform. We have added this information in the discussion section (line 276-280)

15. Line 127-128: The online participation was lower than the offline participation. Wouldn’t this be strange when we are evaluating e-learning?

Authors' response: As mentioned above, the collection data was conducted when students of HUP learnt directly at school and the online learning at HUP stopped. We only conducted online surveys through submitting a google form on the general class group and could not attached on the e-learning platform. Therefore, not every students in the class had access our online questionnaires. This may lead to the low online response rate. The direct survey form was preferred to increase the response rate (line 280-282).

16. Please include any segment necessary as per the above mentioned points

Authors' response: We have added the necessary discussions pointed out by reviewers (line 74-76)

17. Please keep a consistent reference style

Authors' response: We have reviewed and edited to ensure consistency in reference styles.

Thank you very much for your time and consideration. We hope that you are interested in our work.

Your sincerely,

Xuan Thang, Do

---

## [Editor Report · Decision Letter 1]

20 Oct 2022

PONE-D-22-15672R1Factors influencing the E-learning system usage during the COVID-19 pandemic in VietnamPLOS ONE

Dear Dr. Thang,

Thank you for submitting your manuscript to PLOS ONE. After careful consideration, we feel that it has merit but does not fully meet PLOS ONE’s publication criteria as it currently stands. Specifically, we are concerned about some ethical issues such as privacy and confidentiality, which may have introduced some bias e.g. the survey required the students to provide personal identification. Kindly provide information on how you handled these ethical issues. We invite you to submit a revised version of the manuscript that addresses the point raised above.

Please submit your revised manuscript within 15 days. If you will need more time than this to complete your revisions, please reply to this message or contact the journal office at plosone@plos.org. Please include the following items when submitting your revised manuscript:A rebuttal letter that responds to each point raised by the academic editor. You should upload this letter as a separate file labeled 'Response to Reviewers'.A marked-up copy of your manuscript that highlights changes made to the original version. You should upload this as a separate file labeled 'Revised Manuscript with Track Changes'.An unmarked version of your revised paper without tracked changes. You should upload this as a separate file labeled 'Manuscript'.If applicable, we recommend that you deposit your laboratory protocols in protocols.io to enhance the reproducibility of your results. Protocols.io assigns your protocol its own identifier (DOI) so that it can be cited independently in the future. For instructions see: https://journals.plos.org/plosone/s/submission-guidelines#loc-laboratory-protocols. Additionally, PLOS ONE offers an option for publishing peer-reviewed Lab Protocol articles, which describe protocols hosted on protocols.io. Read more information on sharing protocols at https://plos.org/protocols?utm_medium=editorial-email&utm_source=authorletters&utm_campaign=protocols.

We look forward to receiving your revised manuscript.

Kind regards,

Ogochukwu Chinedum Okoye

Academic Editor

PLOS ONE
---

## [Author Response · Author response to Decision Letter 1]

9 Nov 2022

Author's response: To minimize bias, items about personal information would not compulsory (line 108-110). It was optional for respondents to provide personal information. So, it allowed respondents to answer the questionnaire comfortably and to minimize bias.

Besides, the personal information of participants including name, class, phone number and email which can track participants would be coded into ID and only one person could access the original data and this person would not participate in data analysis. The data which was cleaned and coded was imported to IBM SPSS for the purpose of analysis (line 122-125).

---

## [Editor Report · Decision Letter 2]

10 Nov 2022

Factors influencing the E-learning system usage during the COVID-19 pandemic in Vietnam

PONE-D-22-15672R2

Dear Dr Thang,

We’re pleased to inform you that your manuscript has been judged scientifically suitable for publication and will be formally accepted for publication once it meets all outstanding technical requirements.

Kind regards,

Ogochukwu Chinedum Okoye

Academic Editor

PLOS ONE
---

## [Editor Report · Acceptance letter]

24 Nov 2022

PONE-D-22-15672R2 

Factors influencing the E-learning system usage during the COVID-19 pandemic in Vietnam  

Dear Dr. Do:

I'm pleased to inform you that your manuscript has been deemed suitable for publication in PLOS ONE. Congratulations! Your manuscript is now with our production department. 

Kind regards, 

on behalf of

Dr. Ogochukwu Chinedum Okoye 

Academic Editor

PLOS ONE